# Carbon fractions in the world's dead wood

Adam R. Martin[1✉], Grant M. Domke [2], Mahendra Doraisami[1] & Sean C. Thomas[3]

A key uncertainty in quantifying dead wood carbon (C) stocks—which comprise ~8% of total forest C pools globally—is a lack of accurate dead wood C fractions (CFs) that are employed to convert dead woody biomass into C. Most C estimation protocols utilize a default dead wood CF of 50%, but live tree studies suggest this value is an over-estimate. Here, we compile and analyze a global database of dead wood CFs in trees, showing that dead wood CFs average 48.5% across forests, deviating significantly from 50%, and varying systematically among biomes, taxonomic divisions, tissue types, and decay classes. Utilizing data-driven dead wood CFs in tropical forests alone may correct systematic overestimates in dead wood C stocks of ~3.0 Pg C: an estimate approaching nearly the entire dead wood C pool in the temperate forest biome. We provide for the first time, robust empirical dead wood CFs to inform global forest C estimation.

[1] Department of Physical and Environmental Sciences, University of Toronto Scarborough, Scarborough, ON, Canada. [2] USDA Forest Service, Northern Research Station, St. Paul, MN, USA. [3] Institute of Forestry and Conservation, Daniels Faculty of Architecture, Landscape, and Design, University of Toronto, Toronto, ON, Canada. ✉email: adam.martin@utoronto.ca

Forests are a large and dynamic part of the global carbon (C) cycle with estimates indicating an annual average net global forest C sink of 1.1–1.4 Pg C year$^{-1}$ in recent decades[1,2]. Global forest C sinks owe to high net uptake in regenerating forests of ~1.3 Pg C year$^{-1}$; intact forests contribute an additional sink of 0.85–2.4 Pg C year$^{-1}$ [1,2], although recent evidence indicates that the strength of this sink is declining in the tropics[3] and across North America[4]. These sinks are offset by losses of C due to deforestation and forest degradation, particularly in tropical regions where forest loss accounts for ~0.43–1.3 Pg C year$^{-1}$ on average[2,5].

C stocks and fluxes in dead wood—that is, fallen and standing dead trees, branches, and other woody tissues—are a critical component of forest C dynamics. Dead wood accounts for ~8% (or 73 Pg) of total C pool in forests globally[2]. There is wide biogeographic variability in dead wood C stocks and fluxes. For instance, based on 2007 data from Pan et al.[2] (their Table S3), total dead wood C stocks represent ~2.8–11.7% of the total forest C storage across temperate, boreal, and tropical forest biomes. This variability is attributable to differences in primary production, tree mortality, and decomposition rates that are linked with climate and species' wood traits[6–8]. Dead wood C dynamics are also sensitive to fine-scale disturbances such as harvesting, windstorm impacts, wildfires, and pest or pathogen outbreaks (e.g., refs. [9,10]).

Given its importance in the global C cycle, robust methods for quantifying C in dead woody material are critical for estimating forest C stocks and fluxes at multiple scales. However, there remain several critical large sources of uncertainty surrounding estimates of dead wood C stocks and fluxes[11]. One important consideration in estimating dead wood C fluxes that has received limited attention is the proportion of C in dead wood, as it is used to convert dead wood biomass into C stocks[12]. Assessments of dead wood C have most often utilized a single generalized C fraction (CF)—that wood comprises 50% C on a mass/mass basis —when converting woody material mass to C[13–17]. Recent studies have made clear that 50% is a poor approximation of CFs in live trees: the best available global estimate of average live wood CF is ~47.6%, with this estimate ranging from 28 to 65% across biomes, species, and tissue types[18,19]. In live trees, accounting for variability in wood CF corrects major systematic errors in forest C stocks[18–20]. For example, accounting for live wood CF refines existing overestimates of up to 20.1 Pg C in tropical forests[18]. Nevertheless, generalized dead wood CFs have not been obtained for the purposes of global forest C estimation.

Identifying the factors explaining differences in dead woody material CFs has also remained elusive in the absence of data consolidation. Arguably the most important factor driving dead wood CF variability is the decay process, commonly discretized as decay class (DC). There is disagreement in the literature as to the magnitude and direction of changes in CF through decomposition. For instance, studies from temperate and tropical forests have detected little to no change in CFs through decomposition[21–23], others have found increases in CFs[24,25], while others report both decreasing and increasing trends depending on taxonomic divisions (i.e., gymnosperms vs. angiosperms) and tissue type[26–31]. In the absence of a global data compilation and analysis, these contrasting patterns pose a challenge for estimating changes in CFs through wood decay.

Data on CF from live trees also suggest that tissue-specific variability in dead wood CF will be pronounced. Specifically, there is likely to be especially high CFs in bark vs. other tissues, due to their high concentrations of C-rich and recalcitrant compounds such as lignin, suberin, and tannins[32–35]. Finally, the position of dead wood—that is, standing vs. downed—may also influence CFs[12], but hypotheses and findings related to this are mixed with some research suggesting that standing dead wood has higher CFs vs. downed wood[26], while other lines of evidence suggest the opposite[36]. Whether or not these differences are systematic and/or independent of other factors such as biome, species identity, and DC is unclear, as is the relative importance of these factors.

Here, we develop, for the first time, a novel global dataset of 973 dead wood CF observations from 121 species and all forested biomes, to inform forest C estimation and to identify the primary factors determining dead wood CFs in trees. We specifically evaluate: (1) if dead wood CFs differ from (a) the generalized 50% CF estimate commonly employed in forest C estimation, and (b) live wood CFs? As a corollary, we also assess: (2) if live wood CFs predict dead wood CFs within species, (3) if there is systematic and generalizable variability in dead wood CFs across biomes, species, position, and DC, and (4) how do dead wood CFs change through decomposition?

## Results

### Dead wood carbon fractions compared to IPCC protocols and live wood.
Dead wood CFs ranged widely from 29.4 to 60.2% across the compiled dataset, with an average CF estimate of 48.5 ± 0.8% (s.e.). Dead wood CFs are significantly lower than the widely used 50% CF estimate by 1.5% on average (two-sided $z = -6.2$, $p < 0.001$). The average estimated dead wood CFs are also significantly larger than live wood CF that averages 47.2 ± 0.8% ($F_{1, 3392.7} = 67.7$, $p < 0.001$; Fig. 1). Across 63 species with both dead and live wood CFs, the average live wood CFs were significantly related to average dead wood CFs ($r^2 = 0.462$, $p < 0.001$). This relationship differed significantly from a 1:1 relationship across the entire species pool (model slope = 0.7 ± 0.1 (s.e.), linear hypothesis test $p = 0.011$). The intercept of the live–dead wood CF relationship, but not the slope, differed significantly across groups ($p < 0.001$; Supplementary Fig. 1 and Supplementary Tables 5–7). Including taxonomic division-specific intercepts in the linear model (i.e., for angiosperms and gymnosperms individually) explained an additional ~15% of the variation in dead wood CFs (the model $r^2$ when including plant taxonomic division-specific intercept terms = 0.601).

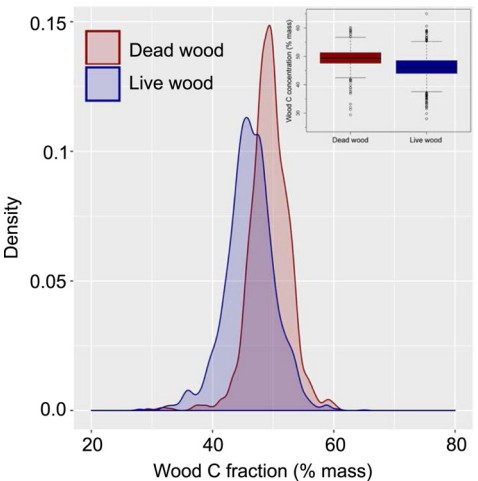

**Fig. 1 Carbon fractions (CF) in dead vs. live wood in a global wood CF database.** Histograms correspond to kernel density estimates fit for CF estimates from dead ($n = 973$) and live wood ($n = 2233$) separately, with the corresponding boxplots in the inset showing medians (solid center lines), 25-75th percentiles (boxes), outliers (points), and 5th and 95th percentiles (whiskers).

**Factors explaining variation in dead wood carbon fractions.** Dead wood CFs varied significantly across biomes, taxonomic divisions (i.e., gymnosperms and angiosperms), tissue types, and DC (analysis of variance (ANOVA) $p < 0.001$; Supplementary Table 1). ANOVA revealed significant interactions between biome and taxonomic division, tissue type, and DC, as well as between position and tissue type (Supplementary Table 1). Variance partitioning indicated that the largest proportion of variability in dead wood CFs was associated with biome (23.1% of variance explained), with systematic and significant differences across all of the biomes represented (Fig. 2, Table 1, and Supplementary Tables 2 and 3). While accounting for all other factors, dead wood CFs in temperate and boreal biomes (49.3 ± 0.8% and 48.8 ± 0.8%, respectively) were ~1.7–2.1% greater in absolute terms than those observed in subtropical/Mediterranean and tropical biomes (46.2 ± 0.8 and 47.2 ± 0.8, respectively; Fig. 2 and Table 1). Tissue type was also a significant factor explaining 18.9% of variability in dead wood CFs (Fig. 2, Table 1, and Supplementary Table 2). Bark, fine tissue, and stem wood showed the largest average dead wood CFs (48.1–48.8%), roots being intermediate (47.8%), and branches showing the lowest average dead wood CF estimates (45.7%; Fig. 2, Table 1, and Supplementary Table 2).

Taxonomic division also explained a significant proportion (7.6%) of the variability in dead wood CFs ($p < 0.001$; Supplementary Tables 2 and 3), with gymnosperm dead wood CFs being 2.0% higher on average compared to angiosperms (Fig. 2 and Table 1). DC explained 8.8% of the variation ($p < 0.001$; Supplementary Tables 2 and 3), with systematic increases in dead wood CFs occurring across DCs 1–3 (average dead wood CF 47.5–48.0%), to DCs 4 and 5 (average dead wood CFs 48.7% and 48.6%, respectively; Fig. 2 and Table 1). There were only slight differences in the CFs of standing vs. downed wood ($p = 0.05$; Fig. 2 and Table 1). Unlike all other factors considered in our analysis, inferences regarding how dead wood position influences CFs did change according to the structure of our variance partitioning analyses (Supplementary Table 2). However, position consistently explained the lowest proportion (≤2.2%) of variation in dead wood CFs. In total, the factors considered here accounted for 58.6% of the variance in dead wood CFs (Supplementary Table 2).

**Dead wood carbon fractions across DCs.** Based on a subset of data that included only species with dead wood CFs from at least four DCs (where $n = 728$ observations across 56 species; Supplementary Table 4), the patterns of change in dead wood CFs with increases in DC varied widely. The majority of species (41 of 56) showed increases in dead wood CF with increasing DC, with species-specific slopes ranging from 0.03 to 1.64; these changes were statistically significant in only five species (i.e., where slope $p ≤ 0.05$, Fig. 3 and Supplementary Table 5). In these 41 species, across DCs 1–5, dead wood CF was predicted to increase on average from 0.15 to 8.2% (Fig. 3). The remaining 15 species showed trends of declining dead wood CF with increasing DC (slope $p ≤ 0.05$ in six instances), with slopes ranging from −0.04 to −4.14% (Fig. 3). The five species with the strongest negative trends (slope $p ≤ 0.002$ in all cases) were all subtropical/ Mediterranean angiosperm species (Fig. 3 and Supplementary Table 5). There were significant differences in slope values between gymnosperms and angiosperms across the entire subset of data ($t_{34.8} = -2.55$, $p = 0.015$, $n = 56$ species; Fig. 3), but these differences were driven by five subtropical angiosperm species with the very largest negative slopes in their wood CF–DC relationship (i.e., slopes <−2.0). When these five species were removed, angiosperms and gymnosperms do not differ in terms of their wood CF–DC relationships ($t_{48.6} = -0.912$, $p = 0.366$, $n = 51$ species; Fig. 3).

**Discussion**

**Dead wood carbon fractions and forest C estimation.** Prominent forest C protocols, namely those of the Intergovernmental Panel on Climate Change (IPCC)[37], are a critical tool in compiling forest C budget data globally, particularly where region- or country-specific data are not available, and support the implementation and monitoring of critical climate change policies and programs. Reducing uncertainty in forest C estimates is therefore a key priority, with the most recent updates to the IPCC protocols updating key default C variables such as tree biomass stocks and growth rates (e.g., Tables 4.4 and 4.7 in ref. [37]). However, the 2019 Refinement to the 2006 IPCC Guidelines for National Greenhouse Gas Inventories[37] included no updates to default dead wood CFs—or default wood CFs in general, despite considerable research on this topic[18]—and instead continued to report a 50% CF as the default estimate for dead wood in temperate forests; there is no IPCC-recommended default CF estimate suggested for dead wood in tropical or subtropical forests.

While deviations in dead wood CFs from the widely used 50% assumption appear small (i.e., 1.5% on average; Fig. 2 and Table 1), this difference ultimately corresponds to a 3% overestimate in dead wood CFs. These findings suggest that existing estimates of dead wood (and hence forest) C stocks are likely overestimated at multiple scales. For instance, at an individual tree scale, in assuming a 50% CF, Domke et al.[15] estimated that a standing dead *Populus tremuloides* stem—a widespread species in North American boreal and temperate forests—of 26.7 cm diameter at breast height and DC 3 would store 22.95 kg of C. Yet when employing our overall average CF (48.5%), temperate biome-specific CF (49.3%), or species-and-DC-specific estimates (mean CF for *P. tremuloides* in DC 3 = 45.8% across $n = 6$ observations), this same tree is calculated to store 22.26, 22.63, or 21.02 kg of C, respectively. Thus, data-driven dead wood CFs correct overestimates of 1.4–8.4%, equating to 0.32–1.93 kg C for a single dead stem.

Scaling this overestimation to forests globally requires the formal integration of our results with models of forest composition. However, to illustrate the potential consequences of our findings at larger scales, inventories that assumed a 50% dead wood CF, reported a total dead wood C stock in tropical forests of 53.6 Pg C in 2007 (corresponding to the total biomass stocks of 107.2 Pg)[2]. However, based on existing data that support an average dead wood CF of 47.16% (±0.79% s.e.) for tropical trees (Table 1), we would estimate tropical forest dead wood C stocks at 49.7–51.4 Pg C (with an average estimate of 50.6 Pg C). This average difference of ~3.0 Pg C is similar to the dead wood C stock in the entire temperate forest biome, which was estimated for the year 2007 as 3.3 Pg C by Pan et al.[2].

There remain several known sources of uncertainty in dead wood C assessments, such as variability in dead wood tissue-density measurements, or limited dead wood inventory data[11]. When compared to these other sources of variability, dead wood CFs can be a minor consideration[11]. Yet, because both inter- and intraspecific variation in wood CFs can be accurately accounted for through accurate measurements[20,38] and representative sampling designs, these overestimates are systematic and easily corrected. Our findings of systematic variation in dead wood CFs across biomes, tissue types, and DCs (and to a lesser extent taxonomic divisions and size classes; Supplementary Table 2) support the calculation and promulgation of generalized dead wood CFs for the purposes of forest C estimation (Table 1). The dead wood CF data compiled here, along with CFs from live

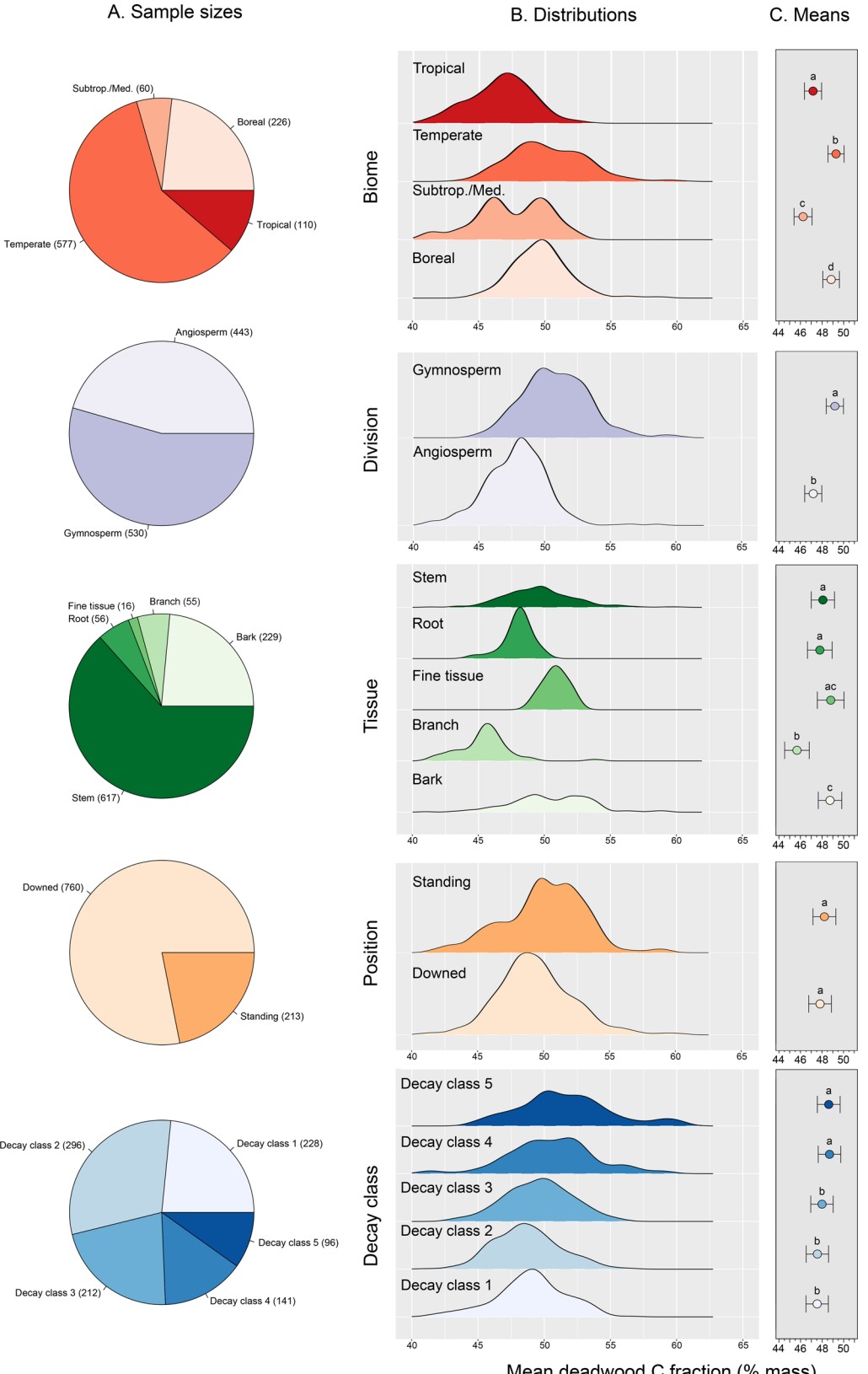

**Fig. 2 Sample sizes, distributions, and mean dead wood carbon fraction (CF) estimates across biomes, taxonomic divisions, tissue type, dead wood position, and decay class.** Panel **A** presents sample sizes. Panel **B** presents kernel density estimates fit to subsets of the dataset (based on the sample sizes presented in **A**). Panel **C** represents least-squares mean estimates (±s.e.) from a linear mixed-effects model fit to the entire dead wood dataset ($n = 973$). Within a given data subset, different letters above mean estimates denote statistically significant differences (at $p < 0.05$) in mean dead wood C estimates.

**Table 1 Generalized mean dead wood carbon fractions (CF) across five different factors.**

| Factors | Variable | Mean CF | S.E. | Lower CI | Upper CI |
|---|---|---|---|---|---|
| Biomes | Boreal | 48.84 | 0.76 | 40.69 | 56.98 |
| | Subtropical/ Medit. | 46.24 | 0.83 | 37.38 | 55.09 |
| | Temperate | 49.29 | 0.74 | 41.29 | 57.28 |
| | Tropical | 47.16 | 0.79 | 38.66 | 55.66 |
| Division | Angiosperm | 47.18 | 0.79 | 44.59 | 49.77 |
| | Gymnosperm | 49.19 | 0.79 | 46.58 | 51.79 |
| Tissues | Branch | 45.67 | 1.14 | 42.13 | 49.21 |
| | Root | 47.79 | 1.14 | 44.25 | 51.33 |
| | Stem | 48.07 | 1.07 | 44.75 | 51.4 |
| | Bark | 48.73 | 1.08 | 45.38 | 52.09 |
| | Fine tissue | 48.8 | 1.23 | 44.97 | 52.63 |
| Position | Downed | 47.81 | 1.05 | 44.32 | 51.31 |
| | Standing | 48.22 | 1.06 | 44.7 | 51.74 |
| Decay class | 1 | 47.53 | 1.03 | 44.16 | 50.9 |
| | 2 | 47.55 | 1.03 | 44.18 | 50.93 |
| | 3 | 47.98 | 1.03 | 44.61 | 51.36 |
| | 4 | 48.68 | 1.04 | 45.28 | 52.08 |
| | 5 | 48.61 | 1.05 | 45.17 | 52.04 |

Mean estimates here were calculated as least-squares means, obtained from five different linear mixed-effects models. Estimates here correspond to data presented in Fig. 2, while linear mixed-effects model diagnostics are presented in Supplementary Table 3.

wood[18], provide a basis for better supported approximations of CFs in trees and wood globally as compared to current IPCC protocols[37].

Our analysis also reveals outstanding sources of uncertainty in dead wood CF data. First, angiosperms remain underrepresented in the literature on dead wood CFs, making up 45.5% of our total data points despite likely constituting over ~95% of tree species globally[39]; this underrepresentative sampling is particularly acute for tropical angiosperms. Second, we uncovered no studies that explicitly quantified the volatile CF in dead wood: low molecular weight C-based compounds that may be lost upon heating of samples, prior to elemental analysis[38,40,41]. Lab methods that fail to capture the volatile fraction can underestimate CFs of live wood by up to 4.7%, with wood volatile fractions averaging ~1.5–2.5% across angiosperms and gymnosperms from different biomes[20,38,40,41]. Third, prior studies show that C-based compounds are leached from downed wood[42], and overall rates and chemistry likely vary among species. To our knowledge, inter- or intraspecific variation in C leaching has not been explored as possible drivers of dead wood CFs. Finally, sapwood and heartwood represent functionally and chemically distinct stem wood tissue types[43,44], although studies have reported both negligible[32] and large differences in their wood CFs[45,46]. Our meta-analysis indicates that heart- and sapwood are not well differentiated in studies on dead wood chemistry, leading to outstanding questions surrounding whether or not stem wood tissue differentiation plays a role in governing dead wood CFs.

These four lines of research therefore represent opportunities for future enquiry, which may help elucidate the reasons for intraspecific differences in live vs. dead wood CFs (Fig. 1 and Supplementary Fig. 1). Specifically, these novel possible avenues of research include observational and experimental studies that: (1) systematically sample and quantify dead wood CFs in tropical angiosperms, (2) quantify and characterize the volatile CF in dead wood across and within species, (3) quantify the rate and extent of C-based compound leaching in dead wood, and its role in influencing inter- and intraspecific variation in dead wood CFs, and (4) isolate and quantify differences in dead wood CFs across sapwood and heartwood across species.

**Factors explaining systematic variation in dead wood carbon fractions.** Our study uncovers the following general patterns in CFs across dead wood globally: (A) lower dead wood CFs in tropical vs. other forest biomes, (B) lower dead wood CFs in angiosperms vs. gymnosperms, and (C) higher dead wood CFs in bark vs. other tissues (Table 1). These results are consistent with studies on live wood CF variability[18,32,33,35,47], and perhaps are not surprising given the statistically significant relationship between dead and live wood CFs observed in a subset of tree species evaluated here (Supplementary Fig. 1). Based on similarities in how dead and live wood CFs vary across and within species, our study indicates that live wood chemical traits (along with their environmental and evolutionary drivers) also play a deterministic, so-called afterlife role (sensu[48]) in driving dead wood C dynamics.

There is considerable variability in patterns of dead wood CF change through decay (Fig. 3), suggesting that multiple mechanisms operate across different species and forest regions. Cellulose and hemicellulose generally decompose more rapidly than lignin[26,49], and lignin has a considerably higher C concentration (~60–70% C mass mass$^{-1}$) than cellulose/ hemicellulose (~40–44% C mass mass$^{-1}$)[50]; thus, CFs would be expected to increase through decomposition as a function of increasing lignin concentrations. Our data on generalized CFs across DCs qualitatively correspond to this expectation (Fig. 2). Quantitatively, assuming an average C concentration of 62.5% for lignin and 41.2% for cellulose, our observed changes in dead wood CFs from DC 1 (47.5%) to DC 5 (48.6%) correspond to an increase in lignin concentrations through decomposition from ~27 to 33% (mass mass$^{-1}$). These approximate changes in lignin concentrations match patterns observed in wood decomposition experiments[49,51,52], consistent with increases in CFs with decomposition in the majority of tree species (Fig. 3).

However, certain species deviate from this pattern and instead show nonsignificant changes or significant declines in CFs through decomposition (Fig. 3). This suggests that there may be mechanisms other than the degradation of cellulose and lignin that drive chemical changes in decomposition globally. One possible mechanism is the import of soil particles and soluble nutrients into dead wood by soil macrofauna—in particular termites[53]—which would reduce dead wood CFs through the decomposition process primarily in tropical and subtropical forests where such patterns were most notable (Fig. 3).

Similarly, there is an expectation that the import of soluble nutrients and particles from soils into woody material should decrease dead wood CFs in downed wood, as compared to standing necromass[26]. Support for this expectation has been observed in temperate and boreal forests, where standing dead trees express significantly greater CFs vs. downed wood (i.e., on the order of ~1.6–2.0%)[26]. This is consistent with our findings of dead wood CFs being higher in standing vs. downed wood, although the magnitude of the average differences in our pooled analysis is lower (~0.4%; Fig. 2).

Finally, our meta-analysis also highlights that the effect of fungal communities on wood decomposition and CFs is still poorly understood and quantified. White-rot fungi, which mainly degrade lignin, are thought to be more prevalent in angiosperms, while brown-rot fungi, which specialize in cellulose degradation, are more prevalent in gymnosperms. This may account for some of the variability in data. However, while removing five subtropical angiosperm species with the largest negative slopes in their wood CF–DC relationship (i.e., slopes <−2.0), angiosperms and gymnosperms did not differ in terms of their wood CF–DC relationships (Fig. 3). Recent studies have suggested that the dichotomy between white-rot and brown-rot fungi is itself not

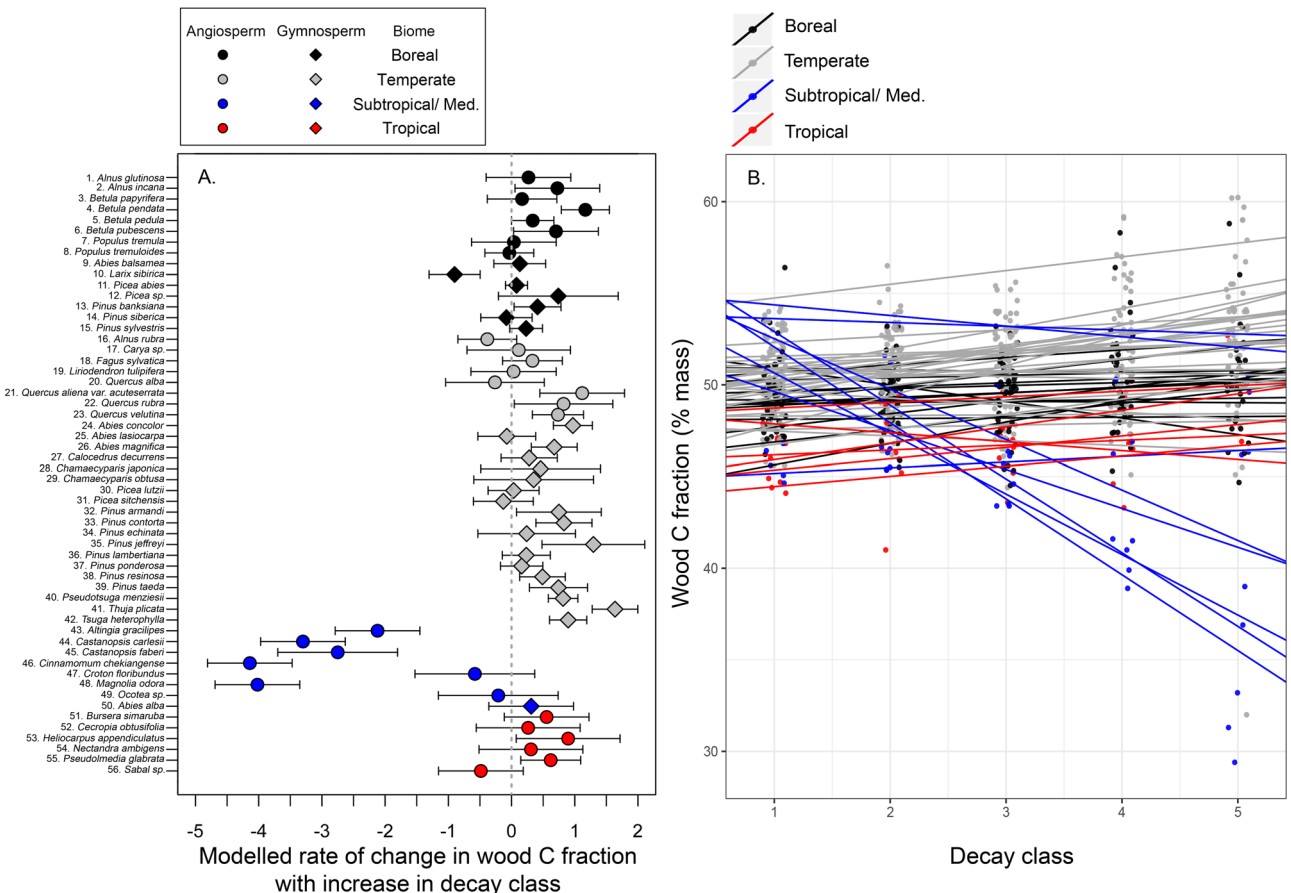

**Fig. 3 Changes in dead wood carbon fractions (CF) as a function of wood decomposition stage.** Panel **A** presents modeled rates of change in dead wood CFs as a function of decay class, which are the slope estimates obtained from a mixed-effects model (±s.e.). Panel **B** presents the species-specific models predicting dead wood CFs as a function of decay class.

strongly supported[54], highlighting the complexity of potential fungal community effects on CF patterns in dead wood. Disentangling how these mechanisms drive variability in CFs through decomposition will likely require detailed experimental studies that evaluate long-term decay patterns[55], account for species differences in wood functional traits[34], incorporate emerging environmental analytical techniques, for example, ref. [56], and test for biochemical changes in wood such as the accumulation of anaerobic metabolic products[57].

At global scales, accurate estimates of CF in dead wood are critical for refining global C budgets, quantifying potential changes in dead wood fluxes under global change scenarios, mechanistically understanding the drivers of decomposition, and predicting how they change in the future. Recent observations of increased tree mortality in multiple forest biomes[58,59] suggest that a synthetic understanding of dead wood chemistry dynamics is especially critical for all of these avenues of forest ecological and global change science.

## Methods

**Literature review.** We built on our existing wood C database[18], which consists of $n = 2,228$ observations of CFs in live wood only, as the basis for dead wood CF consolidation. We first reviewed all peer-reviewed papers that were cited by our previous work, that is, refs. [18–20], for records of dead wood CFs. Then, we searched three peer-reviewed literature databases (Web of Science, Scopus, and Google Scholar) for papers with dead wood CF records, using a suite of primary search terms: coarse woody debris, dead wood, carbon, and wood nutrient. Articles identified by these terms or combinations thereof, as well as papers that cited these

publications, were searched for dead wood CF data. Data compilation was halted at the end of 2019.

The criteria for inclusion broadly followed that of Martin et al.[18], such that only dead wood CF data associated with species identities, tissue-type identities, and decay class (DC) were included in our database. This was done to maximize our sample size, while allowing analysis that was specific enough to inform forest C estimation. For each paper with species- and tissue-specific data, dead wood CF observations were then extracted from text, tables, and figures, with figure-based data extracted using the WebPlotDigitizer software v. 4.2, 2019 (San Francisco, CA, USA). It should be noted that available estimates for dead wood CF were all based on conventionally dried wood samples, and thus exclude volatile C compounds[38]; lab methodologies for the accurate quantification of wood C including volatile constituents continue to be refined[38].

For each observation, we recorded species-specific taxonomy as presented in original publications, which was then adjusted according to the Taxonomic Name Resolution Service v.4.0[60]. Each dead wood CF observation was then classified as belonging to one of four major forested biomes based on its geographical location, including (A) boreal, (B) temperate, (C) subtropical/Mediterranean, and (D) tropical. While these biomes are less specific than more detailed biome delineations (e.g., Whittaker Biomes), we employed them in our analysis in order to be broadly consistent with the biomes used in the IPCC's forest C estimation protocols[37]. Tissue type was recorded as one of the following: (A) bark, (B) stem (inclusive of heartwood and sapwood, which were largely undifferentiated in dead wood CF studies), (C) branch (inclusive of three observations reported as small twigs), (D) roots (large and small, which were by-in-large undifferentiated in dead wood CF publications), and (E) unspecified fine tissue. Two papers reported sampled material as belonging to a category defined as stems and branches, which were classified as stems for analysis here assuming that stems contributed the larger proportion of biomass to these analyses.

Each dead wood CF observation was then categorized according to three primary factors associated with wood decomposition and the related chemical change: (A) DC, (B) position, and (C) size (diameter and length). For the majority of data points (958 of 973), dead wood DC was reported along a conventional 1–5 scale, and was therefore included in our database as published while noting the

DC scale employed. For nine data points where DC was reported as a two-category range (e.g., DC 1–2), the higher DC was used for analysis. In four data points, a multicategory DC was presented (e.g., DC 1–5); therefore, the middle DC estimate was used. In one paper, DC was reported along a 0–5 point scale (where DC 0 is clearly defined in the publication as dead and not live wood); here two data points where dead wood C was reported with a DC of 0 were classified as DC 1. Last, in a subset of papers the number of years since tree death (instead of DC) was reported. In these cases, years since death were converted to DC based on published decay-class transition matrices, for example, ref. [61].

Position was recorded as one of (A) standing, which refers to snags, or (B) downed, which refers to anything sampled from the forest floor. Estimates for the category of suspended woody material were combined with those for snags. A few publications did not differentiate dead wood as being standing vs. downed in the original publication, and instead classified dead wood as standing/ downed. These few cases were classified as downed for analysis here, since there were very few observations in this group (particularly across multiple DCs).

We also sought to integrate dead wood size into our analysis here, but diameter measurements were available for less than 50% of dead wood CF observations, and papers presented a combination of quantitative and categorical measurements. Therefore, diameter measurements were recorded following the original publication, and then categorized into one of seven groups that were chosen to maintain maximum resolution while balancing sample sizes. These diameter groups employed here were: (1) 0.1–1.0, (2) 1.1–2.5, (3) 2.51–5.0, (4) 5.1–10.0, (5) 10.1–20, (6) 20.1–30 cm, and (7) ≥30.1 cm. There are two caveats to these classifications. First, in instances where publications reported size ranges that overlapped two or more of our groups (e.g., one paper reported dead wood as 7–12 cm in diameter), the mid-point of the size range was used to allocate observations into final diameter classes. Second, in cases where dead wood was reported as belonging to undefined categories (e.g. one paper reporting diameter measurements of ≥2.5 cm), all observations from that publication were placed in the next highest diameter group. Length measurements were available only for a small subset of observations, and were recorded as in the original publication and categorized as either (1) 1–100 cm or (2) ≥100 cm.

Our literature-based search was augmented with a structured trait query from the TRY Functional Trait Database[62]. Specifically, we requested records for coarse woody material C concentration (TRY Database trait numbers 841, 868, 1058, and 1153). However, across all of the $n = 202$ data points returned for these traits, 189 data points were associated with live wood estimates from decomposition experiments. Of the remaining 13 dead wood C data points, only 12 were associated with a species identity. However, none of these 12 data points included DC information and were therefore not included in our analysis here. Moreover, we also noted that all of the species represented by these 12 data points were already represented in our dataset (with both dead and live wood C records), indicating that our primary literature review was the most viable avenue for dead wood C data consolidation.

**Data analysis: dead wood CFs vs. live wood CFs and a generalized 50% CF.**
All statistical analyses were performed using R v.3.2.1 (R Foundations for Statistical Computing). First, we utilized a two-tailed $z$ test to evaluate if dead wood CFs across our entire dataset ($n = 973$ observations in total) differed significantly from a 50% CF assumption. Then, two approaches were taken to compare live vs. dead wood CFs. First, we fit a linear mixed-effects model using the "lmer" function in the "lme4" R package[63] to our entire wood CF dataset ($n = 3206$ observations in total from both dead and live wood). In this model, wood CF estimates were predicted as a function of an organism being dead or live (as a fixed effect), while accounting for biome and taxonomic division as random effects. These random effects were incorporated in this model in efforts to better isolate dead vs. live differences since (1) the dead and live CF datasets differ in the number and proportion of observations per biome and taxonomic divisions, and (2) wood CFs vary systematically as a function of biome and taxonomic division; therefore, failing to account for these factors statistically may have biased dead vs. live comparisons. (Note: we also sought to include tissue type as a random effect in this model, although since tissue types are reported differently in live wood than in dead wood, it was not possible to parameterize the model with this random effect.) Based on this model, we then calculated and statistically compared least-squares mean CF estimates for both groups using the "lsmeans" and "difflsmeans" functions in the "lsmeans" R package[64]. Distributions for dead and live wood CF data were presented visually using kernel density estimates calculated in "ggplot2"[65].

Next, we tested if live wood CFs can be used to predict dead wood CFs. Using the subset that included only species with estimates of both, we calculated species-specific mean live wood and dead wood CF estimates, and fit a linear regression to predict dead wood CF from live wood CFs. This linear model was then statistically compared to a 1:1 relationship using the "linearHypothesis" function in the "car" R package[66]. We then included both taxonomic division and division-by-live wood CF interaction terms in this model to evaluate if intercepts and slopes of live–dead wood CF relationship differed among species groups (i.e., gymnosperms vs. angiosperms).

**Data analysis: factors explaining dead wood CFs.** We first used an ANOVA to evaluate if dead wood CFs vary as a function of biome, taxonomic division, tissue type, position, and DC, as well as all two-way interaction terms. We then complemented this ANOVA with a variance partitioning analysis to quantify the proportion of variability in dead wood CFs explained by biome, taxonomic division, tissue type, position, and DC (where $n = 973$ dead wood observations). This analysis followed the methods developed and employed by multiple studies evaluating functional trait variability in plants, for example, refs. [67,68], including our own earlier work on live wood CF variability in trees[18].

Specifically, the variance partitioning analysis entailed fitting a linear mixed-effects model with the "lme" function in the "nlme" R package[69] where all nested levels—namely DC, within position (i.e., standing, down), within tissue, within taxonomic division (i.e., gymnosperms, angiosperm), and within biome)—are entered as sequential random effects, and the overall intercept (or overall mean dead wood CF estimate) is the only estimated fixed effect[67]. We then used the "varcomp" function in the "ape" R package[70] to quantify and partition variation in dead wood CFs across these nested levels. Since certain decisions regarding this nesting structure are subjective, we also performed this analysis with an alternative nesting structure (i.e., position within DC, within tissue, within taxonomic division, and within biome). The variance partitioning analysis was also performed while including size as a factor, but since this necessarily reduced our sample size by over half (to $n = 413$ observations), these results are not presented here.

We then estimated and compared generalized dead wood CF across DCs, positions, tissues, taxonomic divisions, and biomes. Specifically, we fit five linear mixed-effects models wherein one of the five variables (i.e., DC, position, tissue, taxonomic division, and biome) was included as a fixed effect, and the other four variables were included as nested random effects. Based on these five models, we then used the "lsmeans" function to calculate least-squares mean dead wood CFs individually for each DC, position, tissue type, taxonomic division, and biome, and compared them using the "difflsmeans" function.

**Data analysis: changes in dead wood CFs through decomposition.** We evaluated how dead wood CFs change with DC in more specific detail, using a subset of data that included only species with wood C estimates from at least four DCs. For this subset of $n = 56$ species, we then used a linear mixed-effects model to evaluate how wood C changes across DC, and if/how the rate of change differs across species (subset species highlighted in Supplementary Table 4). This analysis entailed using the "lme" function to fit species-specific models predicting dead wood CFs as a function of DC. Specifically, dead wood CFs were predicted as a function of species identity (representing a species-specific intercept) and a species-by-DC interaction term (representing a species-specific slope parameter) as fixed effects, while accounting for biome, taxonomic division, tissue type, and position as random effects. This analysis, which we present and interpret here, treated DC as a continuous variable in order to better facilitate data visualization (i.e., Fig. 3). However, we also performed this analysis while treating DC as an ordinal variable. This analysis yielded the same result interpretations, but was considerably more difficult to present, so is not included here.

## Data availability
The compiled dataset used in our analyses is available through the TRY Functional Trait Database (Dataset ID 755, https://www.try-db.org/TryWeb/Data.php#75), and is available from the corresponding author upon request.

## Code availability
The code used to perform all analyses and generate figures is available upon request to the corresponding author.

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

## Acknowledgements

We thank Mark Harmon for valuable comments on early versions of this paper, as well as his dataset that provided the basis for calculating wood carbon fractions based on lignin concentrations. This research was financially supported by: (1) Discovery Grants from the Natural Science and Engineering Research Council of Canada to both A.R.M. and S.C.T., (2) a University of Toronto Connaught New Researcher Award to A.R.M., (3) a University of Toronto Scarborough International Research Collaboration Fund grant to A.R.M. and G.M.D., (4) the United States Department of Agriculture Forest Service Northern Research Station, and (5) a University of Toronto Scarborough Department of Physical and Environmental Sciences graduate research bursary to M.D. The findings and conclusions in this publication are those of the author(s) and should not be construed to represent any official USDA or U.S. Government determination or policy.

## Author contributions

A.R.M conceived the study, led data compilation and analysis, and wrote the paper; G.M.D. helped write and edit the paper; M.D. contributed to data compilation and helped write and edit the paper; S.C.T. contributed to data compilation and analysis, and helped write and edit the paper.

## Competing interests

The authors declare no competing interests.

## Additional information

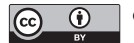

