## [Peer Review File · Nature Communications]

REVIEWER COMMENTS

Reviewer #1 (Remarks to the Author):

I have reviewed the manuscript titled "Carbon fractions in the world's dead wood". Overall I found the manuscript to be clearly written. The topic of CF in dead wood is an important one with global implications. To my knowledge this manuscript would be the first attempt to synthesize the data that exists on CF in dead wood and would therefore add to our understanding of this important pool in forest carbon cycles. I have a few major concerns. In the introduction (and abstract) there are references associated with statements that do not (as far as I could tell) back up the statement being made. I could not tell where specific numbers were coming from as I could not find those values in the cited text (or database). If the authors derived these values themselves they should make that clear. If these values are in the texts then the authors should specify where exactly the values are found so that we can make certain these statements are accurate. In general though these citation issues appear to be confined to the introduction. There are also a few references missing from the manuscript.

There were two topics that I believed needed to be discussed in more detail the first one is the role of VOC content in live tissue CFs. This issue is not mentioned even though the authors directly compare live tree CF to dead tree CF. This issue is important as the methods used to estimate total carbon fraction in tree tissues influence the final CF values that are derived from the analysis. The live tree database that is used in this manuscript consists to a large degree of samples analyzed using oven-dry combustion, this significantly underestimates the amount of carbon in live trees so comparisons between these values and dead wood of the same species could be problematic. The live tree database is also heavily skewed to sapwood samples for the stem wood category, even though most studies that have compared sapwood and heartwood samples have found significant differences between the two wood types. This makes it difficult to interpret how representative the CF values for stem wood are in this study when comparing live to dead tree CFs. More detailed discussion in the methods as to how the CF values for an individual species were derived would help alleviate this to some degree but I did not find information on if CF values were weighted by biomass or not for this study.

The second topic is the role of fungal decay organisms in shifting carbon fractions. We know that generic groups of fungi like brown rot, or white rot fungi differentially feed on lignin or cellulose – thereby shifting the carbon fraction of the resulting decaying wood. This needs to be discussed. In addition to these issues another major issue is with the methods – specifically the data that isn't included here. The authors mention that they attempted to query the TRY Functional Trait Database but that they could not find any observations associated with a species in the single database they looked at – database 868. There are three other databases in the CWD category containing a total of 160 observations, and between 13 – 35 species (range is given in case of overlap between new databases). If those observations are not currently included in this study then they represent a potential increase of 16% in observations, and a 12-31% increase in the number of species represented if we assume those are new species in the missing database. Given the importance of an endeavor such as this I do not believe this manuscript should be published without at least attempting to get that additional data and re-run the analysis.

In this current state I do not believe that this manuscript should be published. Satisfactorily addressing the concerns above and my specific comments below would likely change my evaluation of the manuscript as I believe the topic is important. Specific comments as well as associated line numbers in the manuscript can be found below.

Line 21: The first two references do not measure dead wood - they measure mortality and treat that mortality as an emission rather than tracking the carbon in those dead trees and estimating how much is emitted over time. Western US forests are experiencing increased wildfire - which consumes a lot of dead wood in addition to killing trees - but I would argue that it isn't dead wood contributing to the carbon cycle in that case it is fire. I am also not sure that you can extrapolate from Amazonian, African, and Western US forests to the rest of the globe - you need more direct citations for this

argument. (regarding citation 1-3).

Line 23: I can't find data to back this argument up in that reference - the written version or the database. Where are you getting this information? (regarding citation 4).

Line 33: 1.6 Pg C is actually higher than land-use change emissions estimated by reference 4 for most years.

Line 41: Wasn't it just in Amazonian forests? African forests were relatively stable I believe. (in regards to citation 1).

Line 44: I would call this dead wood - typically it isn't debris until it hits the ground. Fix this for all other references to woody debris when you are actually talking about both standing / down wood.

Line 47: Where are these numbers coming from? That citation doesn't have data on woody decomposition as far as I can tell - and the only table that has info on respiration doesn't have values that match up with that range. Are these values derived somehow? That paper is also missing data from significant portions of the globe so are global estimates meaningful from that study? (in regards to the global fluxes from dead wood).

Line 50: Where is this data coming from? I don't see information on total carbon stocks in dead wood in reference 10, reference 11 is confined to temperate ecosystems and reference 12 is about decay rates. I can't tell how you are coming up with these numbers. (in regards to proportion of C stocks that are dead wood).

Line 63: global estimate - rather than just calling it the best available estimate call it best available global estimate. Scale is important for the reader to think about.

Line 64: I don't believe this is a biomass weighted value (the global CF estimate) - simply the average of values in these data sets (which overlap considerably). That is important as without a biomass weighting any global average CF will be biased if used to calculate total carbon mass. It is also important to note that the majority of the studies in the database (reference 21) used methods that do not accurately estimate live tree tissue carbon fractions because they do not correctly capture the VOC component. Even freeze-drying samples results in some loss of VOC. Adjusting this value for the loss of VOC due to oven-dry combustion would bring it to 49.49 - see: Jones and O'Hara. 2016. The influence of Preparation Method on Measured Carbon Fractions in Tree Tissues. *Tree Physiology* for more information on impact of analytic methods on VOC capture.

It is possible that similar issues exist for dead wood CF values - though the amount of VOC left in dead wood is likely to be much lower than in live wood. Because of that I think it is important to include this topic. If the goal is to improve our understanding of carbon fractions then readers would be aided by more robust coverage (without taking away from your main point).

Line 79: Dominant fungal communities are also likely a cause of this - some fungi preferentially feed on cellulose (brown rot fungi) thereby increasing the remaining CF as lignin is higher in CF, and other fungi feed primarily on lignin (white rot fungi) which would decrease the CF content of the remaining wood. Not sure if there is available information on distribution or dominance of one of these fungal groups over the other but it could be an interesting explanatory variable for determining if CF increases or decreases with decay. (regarding divergence in trends for CF by decay class).

Line 85: Missing some references for wood tissue CF variation. For instance: Jones, Dryw, and Kevin O'Hara. "Variation in Carbon Fraction, Density, and Carbon Density in Conifer Tree Tissues." *Forests* 9, no. 7 (July 18, 2018): 430. <https://doi.org/10.3390/f9070430>. (regarding references for tissue-specific variation)

Line 128: Differences in heartwood and sapwood? Aggregating to stem wood ignores the significant variation that can exist within tree stems. In many cases the heartwood is much more resistant to decay as well so there is good reason to understand the difference in CF values for these two stem wood tissue types. (in regards to only discussing stem wood as a whole). I understand there may not be many examples of separate measurements for heartwood / sapwood but if there are any then it might be worth mentioning. At the very least I hope separate measurements of dead heartwood / sapwood are kept separate – unlike quite a few examples that I have found in the live-tree database on CF where separate heartwood / sapwood samples were simply averaged.

Line 142: That could be a result of more heartwood in larger diameter boles. (in regards to diameter predictive power on CF).

Line 173: It is worth pointing out that a 1.5% deviation from 50% is equivalent to a 3% deviation in the final carbon fraction estimate. All deviations from the 50% estimate are twice as important in the final carbon mass estimate.

Line 174: At what scale? Cousins et al 2015 found higher than 50% CF for all conifers measured. (in regards to C stocks being overestimated).

Line 181: That is comparing annual flux to a change in stock estimates - those aren't equivalent - I also think you don't need this sentence to make the point - you already did that with the previous sentence.

Line 184: And predictions are far more accurate than say for a trait such as wood density, which varies widely and not always predictably throughout a tree, or between trees of the same species within a stand. This is an important aspect of carbon fraction research that most people don't understand – for the most part CF appears to be far more predictable than wood density or even wood volume. It just requires accurate measurements and representative sampling.

Line 196: Missing some references - you will find some of them in: Jones, Dryw, and Kevin O'Hara. "Variation in Carbon Fraction, Density, and Carbon Density in Conifer Tree Tissues." *Forests* 9, no. 7 (July 18, 2018): 430. <https://doi.org/10.3390/f9070430>.

Line 214: This would also depend on which type of fungus infect the wood and which "win" out in decaying the tree faster. Certainly the discussion is an appropriate place to mention the role that fungi can play in the balance of lignin / cellulose.

Line 218: Maybe the biomes give some advantages to different microbial communities - thereby shifting the rate of decay for lignin or cellulose?

Line 222: Also the rapid colonization of fungi once the wood hits the ground.

Line 410: The final figure will hopefully be much larger – very hard to read those species names.

Line 445: That should have been clear in the manuscript - otherwise this is a risky assumption. For instance your reported CF value is not biomass weighted so if nothing is stated about how the CF values were averaged it may be best to assume they are equally weighted. (in regards to the two papers that combined stems and branches)

Line 483: There are multiple data sets for CWD on that website. Specifically: 841 (branch c content - n=83), 1058 (root C content - n=16), 1153 (stem c content - n=61). Many of these databases have multiple species available - why were they not included or requested? These are important data points to add to a paper like this. (in regards to the data query of the TRY Functional Trait Database).

Line 489: Remove the extra then in this sentence.

Line 508: How did you weight the CF values for separate tissue types to come up with a species average? This needs more explanation.

Reviewer #2 (Remarks to the Author):

General comments

On the whole I believe that this manuscript makes an important contribution and documents an important resource for future estimates of dead wood carbon stores. It is generally clearly written and uses appropriate analytical methods.

My main issue is one of tone, which is most problematic in the abstract. Specifically, a number of value-laden modifiers are added in, probably to make the manuscript seem more impactful, but I think they actually detract. As they are not necessary I recommend they be removed.

I agree completely that the time at which a simple 50% carbon concentration was appropriate has long passed. This manuscript makes a very good case for why this is so. But I think it undervalues other uncertainties related to dead wood that frankly are much larger and that is a bit worrisome. Dead wood is still not inventoried as widely as it needs to be, the decay class systems used are poorly defined (if at all), there is little data on density to make the volume to mass conversions, and there is relatively little information on decomposition rates which are necessary to make estimates of fluxes. No one would expect a single manuscript to take all this on, but I think just stating there are some other uncertainties in a general sense does not adequately present the current situation.

Another problem throughout the manuscript is that stores are compared to fluxes. As they have different units this is highly problematical. A similar size of a store and a flux is not meaningful. If the flux is a fraction of the store, say 10% lost per year, then if the store is overestimated by 10%, the amount the flux is overestimated by 10% as well. But when compared to another flux, the dead wood flux is overestimate is considerably less. Using round numbers, if the comparison flux is 10 Pg per year, and the dead wood store is overestimated by 10 Pg and the fraction lost each year is 10%, then the dead wood flux estimate is 1 Pg/y. The point is that the authors need to make appropriate comparisons.

Finally, I agree that more work needs to be done understanding the factors that determine the chemical changes in dead wood. But I think the authors could add a few more details that would guide this effort. For example, is a more systematic sampling needed? One that would be more complete, but also result in a more "balanced" statistical design? Or are there regions or other factors that have been systematically undersampled that should be addressed?

Specific comments (line)

20 I am not sure the word alarming is necessary in this sentence. First it is value laden. Second it is not quantitative in any way. And removing it does not alter the basic meaning of the sentence.

24 However, it has to be noted that this is not the main source of uncertainty, which is related to the volume to mass conversion factor, i.e., density.

27 the meaning of substantial is not clear. I am not sure the word is needed.

29 this is a 3% bias. Is that significant? Again, why not just report the bias and let the reader decide if it is significant?

32 This is an apples to oranges comparison. If I understand this comparison the stores are being compared to a flux: they have different units. This does not make any functional sense.

34 Had the authors actually systematically sampled CF's and then used them I would agree. What they seem to have done is used existing information which is hardly a systematic sampling. I think demonstrating how more empirically based CF's could impact global estimates is an important step. But it is hardly as definitive as being suggested in the abstract.

118 given the irregular design of the dataset I have to wonder if a standard ANOVA would be appropriate.

124 The percent difference could be an absolute or relative difference. Can the authors make this clear? When I do various subtractions I don't get the 1.7-3.1% listed.

152 How can one have a slope for a class-based variable?

180 This is an apples to oranges comparison. Stores and fluxes have different units. How can they be compared?

189 I don't doubt this point at all. But I don't think I would underplay the importance of other issues such as the lack of density values or the ad hoc documentation of decay classes.

201 Well all the dead wood starts as living woody tissues, so it makes sense there is a relationship.

392 As I suspected, the analysis was a bit more involved than the simple ANOVA described under the results. Can the authors provide a bit more detail in the description of the results?

Reviewer #3 (Remarks to the Author):

In their article: "Carbon fractions in the world's dead wood" the authors present a new dataset that addresses an important question in earth system science: what is the concentration of carbon in deadwood? As the authors point out, deadwood accounts for a large and increasingly dynamic fraction of forest C. However, international C budgeting protocols use a blanket value for the C fraction in deadwood: 50%. The authors recently showed, using a similar database and approach, that living trees have a much smaller C fraction that varies widely across forests. Building on this approach, they

examine whether deadwood C fractions differ from 50%, how deadwood C fractions compare to living trees and how deadwood C fractions vary with different features of deadwood, namely, decay class, vertical position, tissue, taxonomic Division (or "Phylum") and biome. In their dataset, deadwood has a C fraction that is intermediate between their estimate for living wood and the assumed value of 50%. Multiplying the average value from their dataset by an estimate of global deadwood biomass results in a much lower value for the size of the global deadwood C pool. Deadwood C fractions varied depending on several commonly measured features of deadwood.

This study has several strengths. It emphasizes an important aspect of the global carbon cycle and presents compelling evidence that deadwood C fractions are lower than had been assumed. The authors also provide an interesting discussion illustrated by very informative figures. Yet, the authors could go further to support their claim (lines 183-184) that deadwood C fraction "biases are systematic and easily corrected" by refocusing their model to estimate the actual deadwood C fraction in forests, not just within their dataset. Doing so may require a more nuanced analysis of factors that contribute to variation in deadwood C. I elaborate on these major points, and suggest some relatively minor changes below.

Major comment 1: How does the average carbon fraction in the dataset scale to global forests?

The authors emphasize the importance of their research by calculating a huge difference in global deadwood C when substituting the average value from their dataset, 48.5% C, for the typical value of 50%. Yet, the majority of their paper makes a persuasive case that deadwood C fractions cannot be represented by a single value (i.e. Table 1, Figure 2, Figure 3, etc). Because C fractions vary systematically by biome, phylum, etc., this information would seem to be necessary to accurately scale up from their dataset (which includes 112 species) to global forests (which may have more than 150,000 species, see FitzJohn, R. G., Pennell, M. W., Zanne, A. E., Stevens, P. F., Tank, D. C., & Cornwell, W. K. (2014). How much of the world is woody?. *Journal of Ecology*, 102(5), 1266-1272.). For example, more than half of their dataset is comprised by conifers, but this Division likely represents less than 1% of all tree species. Similarly, more than half of their observations are from temperate trees, even though temperate forests accounted for only 16.1% of total forest cover between 2000 and 2005 (Hansen, M. C., Stehman, S. V., & Potapov, P. V. (2010). Quantification of global gross forest cover loss. *Proceedings of the National Academy of Sciences*, 107(19), 8650-8655.). Considering that deadwood from conifers and temperate forests had significantly higher C fractions in the analysis of the dataset, it would seem inappropriate to estimate the C content of deadwood globally without reweighting by these and other factors.

Considering that the research objectives, as stated on lines 93-98, do not include estimating deadwood C, the authors could simply drop this estimate from the manuscript. Doing so, I believe, would considerably diminish the potential impact of their work. Instead, I encourage the authors use model not only as a mean of testing hypotheses but also as a rigorous method for more accurately estimating deadwood C. Considering the heterogeneity within the dataset, as illustrated by the very different proportions of variance explained by factors in the whole dataset versus the subset with size measurements (Table S2), an alternative modelling framework may be warranted. An approach that is very well suited to datasets like this is Bayesian hierarchical meta-analysis (e.g. Ogle, K., Pathikonda, S., Sartor, K., Lichstein, J. W., Osnas, J. L., & Pacala, S. W. (2014). A model-based meta-analysis for estimating species-specific wood density and identifying potential sources of variation. *Journal of Ecology*, 102(1), 194-208.). Alternatively, the authors may consider applying Akaike Weights to the set of generalized models they evaluated (e.g. Midolo, G., De Frenne, P., Hölzel, N., & Wellstein, C. (2019). Global patterns of intraspecific leaf trait responses to elevation. *Global change biology*, 25(7), 2485-2498.)

No matter how they choose to proceed with their estimate of deadwood C, it appears that the value for the difference in the Abstract on line 32 differs from that on line 277 and neither has an estimate of uncertainty.

Major Comment 2: How to the factors explain variation in C fraction measurements?

The authors' analyses explain variation in deadwood C fractions attributable to a set of key predictors. Exactly how they code effects varies between analyses, sometimes without a clear rationale. Two issues worth special attention are (1) whether a crossed or nested design is more appropriate and (2) exactly how to code decay class.

(1). The two way ANOVA tested for significant interactions between crossed levels of the factors, while the variance partitioning treated the same factors as nested in a particular order.

When treating the factors as crossed, 4 / 10 possible 2-way interactions were significant (Table S2). The authors discussed one interaction - between biome and decay class - at length (Lines 215-22). Yet, they characterize the results of this analysis as yielding interaction terms that "were largely non-significant predictors of dead wood CFs" (lines 541-542). This characterization, which is provided as a justification for excluding the interactions from one set of analyses, seems at odds with both the statistical results and their biological interpretation.

With respect to the nested ANOVA, the authors describe a specific nesting pattern for the factors. However, it was not clear why DC should nest within Position versus the other way around. The order for nesting these factors and others may be more important than the text implies. Table 1 in Box 1 from Messier et al (cited by the authors) derives the relationship between the order of nesting, the mean squares and the magnitude of different variance components. However, it does not show that variance estimates are commutative over the order of the nested levels. Rather, their derivation seems to suggest that the expected mean squares depend on the order of nesting and the number of factors within a level. It may be worthwhile to reanalyze the data with alternative nesting orders. Also, the Messier et al. analysis included species as one of the factor nesting levels, which would seem to provide a better justification for treating tissue, decay class and position as nested for the purposes of this analysis.

No matter how the authors choose to code their factor effects, nested versus crossed, a full cross tabulation of their data would make it easier to understand the structure.

(2). In the two-way ANOVA (e.g. Table S1), the authors treat DC as a categorical, while in the analysis of covariance (e.g. Figures S1-2), the authors treat DC as interval-valued. I generally think of Decay Class as ordinal (Oberle, B., Lee, M. R., Myers, J. A., Osazuwa-Peters, O. L., Spasojevic, M. J., Walton, M. L., ... & Zanne, A. E. (2020). Accurate forest projections require long-term wood decay experiments because plant trait effects change through time. *Global Change Biology*, 26(2), 864-875) but have also analyzed Decay Class as categorical (Oberle, B., Ogle, K., Zanne, A. E., & Woodall, C. W. (2018). When a tree falls: Controls on wood decay predict standing dead tree fall and new risks in changing forests. *PloS one*, 13(5), e0196712.). I have some reservations about treating DC as a continuous variable. It seems unlikely to me that the difference between DC1 and DC2 are identical to those between DC3 and DC4. DC 3 is much more abundant in typical forests, implying that deadwood has a much longer residence time in this state. For this reason, I think it may be a stretch to convert deadwood ages into decay classes (e.g. Lines 45-457). Furthermore, I would take the estimates of slopes and intercepts from the Analysis of Covariance (e.g. Figure 3 b. and Figure S2) with a grain of salt. Not only does the X axis have an arbitrary scale (Figure 3b), but also, the coordinates are species-level means that aggregate different amounts of variability in Figure S2). Because decay class contributes to variation in dead wood C fractions along the Y axis, but not live wood C fractions along the X axis, error variation would tend to dampen the slope of the estimated relationship due to aggregation and regression, rather than any biological process.

Minor Comments:

L 72-73: This sentence is somewhat confusing. Is it necessary?

L 79: By convention, botanists use "Division" for this rank, not "Phylum". Conifers is the best description for the Division Coniferophyta, which includes all of the species in the study. Elsewhere, the authors use the term "Gymnosperms" which is broader and includes other Divisions like Ginkgophyta.

L 81: Is "generic" necessary?

L148-150: I found the wording of this sentence confusing.

L 202-213: Fascinating discussion.

L 226-227: This seems to imply a DC x Position interaction that was not supported by the two-way ANOVA.

L 244: "with" wish

L 392: I could not find Equation 1 in the text or supplement.

L 437: What were the criteria for identifying Biome? Stating them could help explain why "Picea sp." is classified as boreal even though two species of Picea are classified as temperate.

L 450-453: The rules for imputing DC seemed a bit arbitrary. What is the impact of censoring these observations?

L 479-483: Is this method relevant?

L 526: i.e. standing, down

Figure S1: This figure does not seem necessary

~Brad Oberle

REVIEWERS' COMMENTS

Reviewer #1 (Remarks to the Author):

The authors have done an excellent job responding to my comments. I suggest clarifying that stemwood tissue types are not well studied and a casual reader needs to be informed that CF fractions can be significantly different between tissue types within a species or at least not come away thinking that sapwood and heartwood are always equal across all species. For this paper though since it is focused on dead wood I think a sentence or two about how stemwood tissue types are not differentiated in the data and therefore could be an area of future study would be useful. I think those sentences could be added to the paragraph ending on line 219 fairly easily. If the authors pursue the volatile CF angle on deadwood in a future study it would be helpful to look at stemwood tissue types as well as they may find something interesting there in terms of decay patterns and their relationships with CF values.

I congratulate the authors on this useful addition to the literature and believe it will help further out understanding of carbon science. I look forward to future publications from the authors!

Sincerely,

Dryw Jones

Reviewer #2 (Remarks to the Author):

General comments

I have reviewed the author's response to reviewers and the revised manuscript. Overall I am satisfied with both the responses and the revisions. The manuscript is much improved. There are several minor comments below that should be easy to address. The only major comment may also be easy to address, but might involve a bit more writing.

My major comment is related to the discussion about more need to study CF's in dead trees. I completely agree, but I think the authors could do a bit more in terms of making a recommendation other than to do more. I believe what is needed is a more systematic sampling that moves us out of the "a hoc" stage of this particular science. Moreover, it needs to be global in scope because more work in locations with lots of past work might not address the problem at the scale in needs to be solved. The authors note there have been biases regarding the taxa that have been sampled. That could be addressed by a more global sampling design that covers underrepresented locations and taxa. They also note that some variables such as size cannot be analyzed because size was not reported. This could be addressed by proposing that a standard set of variables be determined including species, decay class, position, size, and other factors they consider important (dominant decomposer organisms present?). These recommendations would address other key uncertainties related to dead wood stores and dynamics. For example, there is no reason that density could not be determined on the samples used to determine carbon concentration. And while I understand these recommendations would have to be limited to a few sentences, they still would be extremely important to present.

Specific comments (line)

55 It is odd to not see wildfires on this list of disturbances. Clearly wildfires are important. So why leave them off the list?

74 I have never seen decay classes referred to as wood decay classes. Not sure the wood adds anything and if decay classes need a modifier it should be CWD decay classes.

107 I have to wonder if an r^2 of 0.462 is a "strong" relationship. If that is strong then what would an

r^2 of >0.9 indicate? Hyper-strong? Perhaps the relationship observed is moderate or moderately strong? This way stronger relationships don't need to be given odd modifiers.

215 Another possible issue is the fact that CWD samples often have their densities determined using water displacement. If these samples are reused to determine the carbon concentration then some of the carbon leached out. It is not clear how much or if a leachable fractions are more or less concentrated in terms of carbon. This might be mentioned in a sentence.

226 Again, why was this a strong relationship?

Reviewer #3 (Remarks to the Author):

In their revised manuscript, the authors went to considerable length to address concerns that I raised about the original submission, as well as those raised by the other referees. With respect to my major concerns about how they scaled up their estimate of C fractions and made decisions about how to code certain factors, I am largely satisfied that the major conclusions are adequately supported by robust analyses.

With respect to the issue of scaling, I believe that the targeted estimates provided by the authors are more likely to be accurate and reinforce the main point of the paper: that systematic differences between regions and major groups may bias estimates of deadwood C. I anticipate an interesting follow-up which applies this model more rigorously.

With respect to the coding scheme for different predictors, I appreciate the considerable effort the authors took to address the impacts of alternative schemes. For the nesting order, I agree that the apparent sensitivity of variance partitioning to the order of variables when including diameter justifies playing down the diameter dataset. That said, the results should point out that the importance of position (as represented by variance component) changes by a factor of 10 depending on the ordering. Although that does not change the rank order of importance, it certainly changes the interpretation. Otherwise, I appreciate the analyses of DC class as ordinal and hope that readers get to the caveat in the methods rather than attempt to replicate what I believe (and reviewer 2 agrees) to be a problematic way of analyzing decay class data.

Given those major changes and others in response to the comments by reviewers 1 and 2, I think this manuscript is very much improved and likely to make a positive contribution to the field.

~Brad Oberle

REVIEWERS' COMMENTS

Reviewer #1 (Remarks to the Author):

The authors have done an excellent job responding to my comments. I suggest clarifying that stemwood tissue types are not well studied and a casual reader needs to be informed that CF fractions can be significantly different between tissue types within a species or at least not come away thinking that sapwood and heartwood are always equal across all species. For this paper though since it is focused on dead wood I think a sentence or two about how stemwood tissue types are not differentiated in the data and therefore could be an area of future study would be useful. I think those sentences could be added to the paragraph ending on line 219 fairly easily. If the authors pursue the volatile CF angle on deadwood in a future study it would be helpful to look at stemwood tissue types as well as they may find something interesting there in terms of decay patterns and their relationships with CF values.

Agreed. We have now included a note that differences in CFs across sapwood and heartwood is currently 1) not particularly well characterized in dead wood, but 2) represents a central avenue for future studies on the topic. While the Reviewer recommends this be included a single sentence (or two), this important point has been integrated within a larger “future directions” section suggested by Reviewer 2 below (Page 8, Lines 223-228).

I congratulate the authors on this useful addition to the literature and believe it will help further out understanding of carbon science. I look forward to future publications from the authors!

We kindly thank Dr. Jones for the continued attention, support, and exceptional insights that he afforded to our manuscript.

Sincerely,

Dryw Jones

Reviewer #2 (Remarks to the Author):

General comments

I have reviewed the author's response to reviewers and the revised manuscript. Overall I am satisfied with both the responses and the revisions. The manuscript is much improved. There are several minor comments below that should be easy to address. The only major comment may also be easy to address, but might involve a bit more writing.

We thank the reviewer for their continued attention and support of our paper.

My major comment is related to the discussion about more need to study CF's in dead trees. I completely agree, but I think the authors could do a bit more in terms of making a recommendation other than to do more. I believe what is needed is a more systematic sampling that moves us out of the "a hoc" stage of this particular science. Moreover, it needs to be global in scope because more work in locations with lots of past work might not address the problem at the scale in needs to be solved. The authors note there have been biases regarding the taxa that have been sampled. That could be addressed by a more global sampling design that covers underrepresented locations and taxa. They also note that some variables such as size cannot be analyzed because size was not reported. This could be addressed by proposing that a standard set of variables be determined including species, decay class, position, size, and other factors they consider important (dominant decomposer organisms present?). These recommendations would address other key uncertainties related to dead wood stores and dynamics. For example, there is no reason that density could not be determined on the samples used to determine carbon concentration. And while I understand these recommendations would have to be limited to a few sentences, they still would be extremely important to present.

Agreed. We have now included a more in-depth "future directions" section in our paper, which is directly informed by our previous (and now edited) section noting outstanding deficiencies in currently published dead wood CF data (Page 8, Lines 211-228). Specifically, based on this comment, we now explicitly identify four lines of future work that our meta-analysis reveals as being viable and informative. This new section specifically reads as follows (Page 8-9, Lines 229-237):

"These four lines of research therefore represent opportunities for future enquiry, which may help elucidate the reasons for intra-specific differences in live vs. dead wood CFs (Figures 1, S1). Specifically, these novel possible avenues of research include observational and experimental studies that: 1) systematically sample and quantify dead wood CFs in tropical angiosperms; 2) quantify and characterize the volatile CF in dead wood across and within species; 3) quantify the rate and extent of C-based compound leaching in dead wood, and its role in influencing inter- and intraspecific variation in dead wood CFs; and 4) isolate and quantify differences in dead wood CFs across sapwood and heartwood across species."

Specific comments (line)

55 It is odd to not see wildfires on this list of disturbances. Clearly wildfires are important. So why leave them off the list?

Agreed. “Wildfires” has been added to this list of localized disturbances governing dead wood C dynamics.

74 I have never seen decay classes referred to as wood decay classes. Not sure the wood adds anything and if decay classes need a modifier it should be CWD decay classes.

Agreed. The term “wood” has been removed.

107 I have to wonder if an r^2 of 0.462 is a “strong” relationship. If that is strong then what would an r^2 of >0.9 indicate? Hyper-strong? Perhaps the relationship observed is moderate or moderately strong? This way stronger relationships don’t need to be given odd modifiers.

Agreed. This is a good point. We have removed our modifier “strongly” from this sentence, and allow the r^2 value to speak for itself.

215 Another possible issue is the fact that CWD samples often have their densities determined using water displacement. If these samples are reused to determine the carbon concentration then some of the carbon leached out. It is not clear how much or if a leachable fraction is more or less concentrated in terms of carbon. This might be mentioned in a sentence.

Interesting idea. To our knowledge, the idea of C leaching from wood has primarily been explored in the area of wood products and processing (e.g., Svensson et al., 2013, Water and Environment Journal). However, this is a possibility. We have added a short sentence to note this (Page 8, Lines 22-223), and this line of enquiry has also been noted as a highly informative “future research” direction (Pages 8-9, Lines 234-236).

226 Again, why was this a strong relationship?

Agreed. This has been changed to “statistically significant relationship.”

Reviewer #3 (Remarks to the Author):

In their revised manuscript, the authors went to considerable length to address concerns that I raised about the original submission, as well as those raised by the other referees. With respect to my major concerns about how they scaled up their estimate of C fractions and made decisions about how to code certain factors, I am largely satisfied that the major conclusions are adequately supported by robust analyses.

We sincerely thank Dr. Oberle for his continued attention and support of our manuscript.

With respect to the issue of scaling, I believe that the targeted estimates provided by the authors are more likely to be accurate and reinforce the main point of the paper: that systematic differences between regions and major groups may bias estimates of deadwood C. I anticipate an interesting follow-up which applies this model more rigorously.

We thank Dr. Oberle for raising these issues of scaling in our original manuscript, and likewise, look forward to exploring this avenue of research in more detail.

With respect to the coding scheme for different predictors, I appreciate the considerable effort the authors took to address the impacts of alternative schemes. For the nesting order, I agree that the apparent sensitivity of variance partitioning to the order of variables when including diameter justifies playing down the diameter dataset. That said, the results should point out that the importance of position (as represented by variance component) changes by a factor of 10 depending on the ordering. Although that does not change the rank order of importance, it certainly changes the interpretation.

Agreed. We have now made a note in our results that 1) unlike the other factors, inferences regarding the influence of position on dead wood CFs did depend on the nesting structure, but that 2) position clearly and consistently explains the lowest proportion of variation in dead wood CFs (Page 5, Lines 137-140).

Otherwise, I appreciate the analyses of DC class as ordinal and hope that readers get to the caveat in the methods rather than attempt to replicate what I believe (and reviewer 2 agrees) to be a problematic way of analyzing decay class data.

Given those major changes and others in response to the comments by reviewers 1 and 2, I think this manuscript is very much improved and likely to make a positive contribution to the field.

~Brad Oberle

Again, we wish to express our sincere gratitude to Dr. Oberle for the time, effort, and detailed thoughts he allocated to our manuscript.